# Thrombosis and thrombocytopenia after vaccination against and infection with SARS-CoV-2 in Catalonia, Spain

Population-based studies can provide important evidence on the safety of COVID-19 vaccines. Here we compare rates of thrombosis and thrombocytopenia following vaccination against SARS-CoV-2 with the background (expected) rates in the general population. In addition, we compare the rates of the same adverse events among persons infected with SARS-CoV-2 with background rates. Primary care and linked hospital data from Catalonia, Spain informed the study, with participants vaccinated with BNT162b2 or ChAdOx1 (27/12/2020-23/06/2021), COVID-19 cases (01/09/2020-23/06/2021) or present in the database as of 01/01/2017. We included 2,021,366 BNT162b2 (1,327,031 with 2 doses), 592,408 ChAdOx1, 174,556 COVID-19 cases, and 4,573,494 background participants. Standardised incidence ratios for venous thromboembolism were 1.18 (95% CI 1.06-1.32) and 0.92 (0.81-1.05) after first- and second dose BNT162b2, and 0.92 (0.71-1.18) after first dose ChAdOx1. The standardised incidence ratio for venous thromboembolism in COVID-19 was 10.19 (9.43-11.02). Standardised incidence ratios for arterial thromboembolism were 1.02 (0.95-1.09) and 1.04 (0.97-1.12) after first- and second dose BNT162b2, 1.06 (0.91-1.23) after first-dose ChAdOx1 and 4.13 (3.83-4.45) for COVID-19. Standardised incidence ratios for thrombocytopenia were 1.49 (1.43-1.54) and 1.40 (1.35-1.45) after first- and second dose BNT162b2, 1.28 (1.19-1.38) after first-dose ChAdOx1 and 4.59 (4.41- 4.77) for COVID-19. While rates of thrombosis with thrombocytopenia were generally similar to background rates, the standardised incidence ratio for pulmonary embolism with thrombocytopenia after first-dose BNT162b2 was 1.70 (1.11-2.61). These findings suggest that the safety profiles of BNT162b2 and ChAdOx1 are similar, with rates of adverse events seen after vaccination typically similar to background rates. Meanwhile, rates of adverse events are much increased for COVID-19 cases further underlining the importance of vaccination.

The advent of the coronavirus disease (COVID-19) vaccines has raised hopes of an end of the COVID-19 pandemic. As of June 2021, the European Medicines Agency (EMA) has now authorised four vaccines against the severe acute respiratory syndrome coronavirus 2 (SARS-CoV-2), the virus that causes COVID-19[1]. Of these, two are mRNA vaccines: BNT162b2 mRNA (manufactured by Pfizer-BioNtech, approved on 21 December 2020) and mRNA-1273 (Moderna, 6 January 2021); and two are adenovirus-based vaccines: ChAdOx1 nCoV-19

e-mail: daniel.prietoalhambra@ndorms.ox.ac.uk; tduarte@idiapjgol.org

(Oxford–AstraZeneca, 29 January 2021), from now on ChAdOx1, and Ad.26.COV2.S (Janssen, 11 March 2021). These vaccines have shown a high degree of efficacy against symptomatic COVID-19 in randomised trials (70-95%);[2–5] and an increasing body of real-world evidence shows that they are highly effective in reducing infections, hospitalisations, and deaths[6,7]. However, concerns have been raised regarding their safety.

In March 2021, several European countries paused and/or suspended vaccination with ChAdOx1 after spontaneous reports of unusual thromboembolic events associated with thrombocytopenia among ChAdOx1 recipients[8]. As of 21 March, the EMA had reported 62 cases of cerebral venous sinus thrombosis (CVST) and 24 cases of splanchnic venous thrombosis (SVT) in the European Union and the United Kingdom in relation to ChAdOx1 (with 25 million doses administered to that date)[9]. Similarly, in the United States, 17 thromboembolic events with thrombocytopenia had been reported as of April 2021 in relation to Ad.26.COV2.S vaccine (with nearly 8 million doses administered)[10]. Subsequently, two population-based studies found increased rates of thromboembolic events among people vaccinated with one dose of ChAdOx1 when compared to expected rates in the general population[11,12]. Although far less reported, the occurrence of thrombocytopenia has also been seen among some recipients of mRNA vaccines[13]. With millions of people already having been vaccinated against COVID-19, population-based studies can provide important evidence on the safety of COVID-19 vaccines.

In this work we aimed to describe the incidence rates of thrombosis and thrombocytopenia after vaccination with BNT162b2 (first and second-dose) and ChAdOx1 (first-dose) and to compare these with the rates seen among the general population before the COVID-19 pandemic. In addition, we set out to estimate incidence rates of the same events for persons with COVID-19 to help contextualise our findings.

## Results

We included 2,613,774 people vaccinated against SARS-CoV-2 (2,021,366 with a first-dose of BNT162b2 and 592,408 with a first-dose of ChAdOx1) in the study. A total of 1,327,031 second doses were observed for BNT162b2 (65.7%), with 74% of these seen at exactly 21 days after the first, 1% seen on days 18–20 and the remaining 24% seen between days 22–28. In addition, we included 174,556 COVID-19 cases and 4,573,494 people in the general population cohort. People vaccinated with BNT162b2 were on average younger than those vaccinated with ChAdOx1, see Table 1, but with a much wider age distribution, Fig. 1. Indeed, in accordance with national guidelines, the majority of those vaccinated with ChAdOx1 (72.2%) were aged 60–69 years. Vaccinations over time are shown in Fig. 2, stratified by age group. Recipients of both BNT162b2 and ChAdOx1 were typically older than COVID-19 cases and those in the general population cohort. Supplementary Table 1 shows participants' characteristics by cohort and age category.

### Venous thromboembolism events

In the first 21 days following a first-dose of BNT162b2, we observed 182 instances of deep vein thrombosis (DVT), which compared with 176 expected events (standardised incidence rate [SIR]: 1.03 [95% CI 0.89–1.19]), and 154 occurrences of pulmonary embolism (PE), which compared with 123 expected events (1.25 [1.07-1.46]). After a second-dose of BNT162b2, however, 130 instances of DVT were observed which compared with 162 expected (0.80 [0.67–0.95]), while 116 PE were seen which compared with 115 expected (1.00 [0.84–1.20]). For the first-dose ChAdOx1 cohort, we saw 39 DVT, which compared with 43 expected (0.89 [0.65–1.22]), and 24 PE, which compared with 30 expected (0.78 [0.52–1.16]). Instances of splanchnic venous thrombosis were in line with expected rates for all the vaccinated cohorts, but were higher for the COVID-19 diagnosed cohort (2.64 [1.53–4.55]).

Overall, rates of venous thromboembolism (VTE, a composite of DVT and PE) were higher than expected after first-dose of BNT162b2 (1.18 [1.06–1.32]) but in line with expected rates after second-dose of BNT162b2 (0.92 [0.81–1.05]) and first-dose of ChAdOx1 (0.92 [0.81–1.05]). In comparison, while 61 occurrences of VTE would have been expected among COVID-19 cases in the absence of the disease, we saw 630 instances of VTE among this cohort (10.19 [9.43–11.02]), Table 2 and Fig. 3.

Incidence rates by age are shown in Fig. 4, with incidence rates increasing with age but broadly similar among those vaccinated as for the general population. Incidence rate ratios (IRR) by age group are shown in Fig. 5, with risks much increased among COVID-19 cases compared to the background population for all the age groups.

### Arterial thromboembolism events

Rates of arterial thromboembolism (ATE, a composite of myocardial infarction and ischaemic stroke) after vaccination were similar to expected rates for both vaccines. We observed 793 ATE after a first dose of BNT162b2, which compared with 780 expected (SIR: 1.02 [0.95–1.09]), 774 after a second-dose of BNT162b2, which compared with 741 expected (1.04 [0.97–1.12]), and 178 after a first dose of ChAdOx1, which compared with 168 expected (1.06 [0.91–1.23]). Conversely, rates of ATE for COVID-19 cases were much higher than expected, with 683 ATE seen among COVID-19 cases, which compared with 165 expected (4.13 [3.83–4.45]).

Incidence rates of ATE also increased by age (Fig. 4). However, when stratifying risks of ATE by age group, we observed higher than expected IRRs among first-dose BNT162b recipients aged 50–59 (IRR: 1.33 [1.11–1.58]) and 60-69 years (1.54 [1.09–2.13]), and second-dose BNT162b2 recipients aged 70–79 years (1.19 [1.06–1.33]) compared to the general population. These risks where much increased among COVID-19 cases for all the age groups.

### Thrombocytopenia

We observed more cases of thrombocytopenia than expected following both first- and second-dose of BNT162b2 and first-dose of ChAdOx1. Following a first-dose of BNT162b2, we observed 3186 cases, which compared with 2145 expected (SIR: 1.49 [1.43–1.54]). After second-dose, 2749 cases were seen, which compared with 1963 expected (1.40 [1.35–1.45]). Following a first-dose of ChAdOx1, we observed 754 cases, which compared with 588 expected (1.28 [1.19–1.38]). Among COVID-19 cases, 2476 cases were seen which compared to 560 expected (4.59 [4.41–4.77]).

When stratifying by age group, we found higher than background (expected) IRRs for thrombocytopenia after first-dose of BNT162b2 among individuals aged 50 years and older, Fig. 5. For example, for individuals aged 60–69 years, IRRs were 1.50 (1.25–1.80). Conversely, for first-dose of ChAdOx1, IRRs remained significant only for individuals aged 60–69 years (IRR: 1.30 [1.20–1.41]), although there was greater uncertainty around estimates for this vaccine among younger age groups.

Diagnoses of immune thrombocytopenia were though less or equal to expected for both vaccines. In total, 97 cases were seen after a first-dose of BNT162b2, which compared to 94 expected (SIR: 1.03 [0.84–1.26]), 61 after a second-dose of BNT162b2, which compared to 89 expected (0.69 [0.53–0.88]), and 12 after a first-dose of ChAdOx1, which compared to 24 expected (0.48 [0.27–0.85]). Conversely, 292 cases were seen among those COVID-19 cases, which compared with 22 expected (13.29 [11.85–14.91]).

### Thrombosis with thrombocytopenia

Rates of deep vein thrombosis with concomitant thrombocytopenia were in line with expected rates in all the vaccinated cohorts. Conversely, rates of pulmonary embolism with thrombocytopenia were

**Table 1 | Characteristics of study participants**

| | General population | BNT162b2 first-dose | BNT162b2 second-dose | ChAdOx1 first-dose | COVID-19 cases |
|---|---|---|---|---|---|
| *N* | 4,573,494 | 2,021,366 | 1,327,031 | 592,408 | 174,556 |
| Age | 48 [37–63] | 55 [45–75] | 70 [54–79] | 62 [59–65] | 47 [35–61] |
| Age: 20–29 | 572,054 (12.5%) | 41,288 (2.0%) | 32,011 (2.4%) | 25,029 (4.2%) | 27,629 (15.8%) |
| Age: 30–39 | 863,539 (18.9%) | 183,972 (9.1%) | 35,774 (2.7%) | 34,221 (5.8%) | 29,023 (16.6%) |
| Age: 40–49 | 979,378 (21.4%) | 527,768 (26.1%) | 105,665 (8.0%) | 48,145 (8.1%) | 38,071 (21.8%) |
| Age: 50–59 | 776,128 (17.0%) | 482,366 (23.9%) | 398,367 (30.0%) | 57,038 (9.6%) | 31,446 (18.0%) |
| Age: 60–69 | 599,531 (13.1%) | 69,304 (3.4%) | 59,135 (4.5%) | 427,885 (72.2%) | 19,023 (10.9%) |
| Age: 70–79 | 428,023 (9.4%) | 401,584 (19.9%) | 389,924 (29.4%) | 86 (0.0%) | 13,797 (7.9%) |
| Age: 80 or older | 354,841 (7.8%) | 315,084 (15.6%) | 306,155 (23.1%) | <5 | 15,567 (8.9%) |
| Sex: Male | 2,233,092 (48.8%) | 928,908 (46.0%) | 569,285 (42.9%) | 270,395 (45.6%) | 80,723 (46.2%) |
| Years of prior observation time | 11.0 [11.0–11.0] | 15.3 [15.2–15.4] | 15.3 [15.2–15.4] | 15.3 [15.2–15.3] | 14.8 [14.7–15.1] |
| **Comorbidities** | | | | | |
| Autoimmune disease | 78,636 (1.7%) | 50,255 (2.5%) | 41,345 (3.1%) | 14,485 (2.4%) | 3843 (2.2%) |
| Antiphospholipid syndrome | 988 (0.0%) | 1404 (0.1%) | 992 (0.1%) | 368 (0.1%) | 104 (0.1%) |
| Thrombophilia | 2708 (0.1%) | 2922 (0.1%) | 1939 (0.1%) | 679 (0.1%) | 264 (0.2%) |
| Asthma | 264,435 (5.8%) | 140,284 (6.9%) | 92,296 (7.0%) | 36,297 (6.1%) | 13,219 (7.6%) |
| Atrial fibrillation | 137,259 (3.0%) | 111,726 (5.5%) | 105,525 (8.0%) | 13,654 (2.3%) | 7339 (4.2%) |
| Malignant neoplastic disease | 338,633 (7.4%) | 243,671 (12.1%) | 219,790 (16.6%) | 60,079 (10.1%) | 15,408 (8.8%) |
| Diabetes mellitus | 464,169 (10.1%) | 286,446 (14.2%) | 245,320 (18.5%) | 80,496 (13.6%) | 20,803 (11.9%) |
| Obesity | 851,541 (18.6%) | 520,035 (25.7%) | 407,283 (30.7%) | 160,914 (27.2%) | 42,536 (24.4%) |
| Heart disease | 566,359 (12.4%) | 408,181 (20.2%) | 366,302 (27.6%) | 87,417 (14.8%) | 26,193 (15.0%) |
| Hypertensive disorder | 1,138,877 (24.9%) | 709,293 (35.1%) | 632,457 (47.7%) | 200,989 (33.9%) | 43,554 (25.0%) |
| Renal impairment | 229,003 (5.0%) | 195,554 (9.7%) | 184,698 (13.9%) | 21,368 (3.6%) | 12,887 (7.4%) |
| COPD | 165,700 (3.6%) | 109,761 (5.4%) | 100,699 (7.6%) | 30,281 (5.1%) | 7265 (4.2%) |
| Dementia | 72,171 (1.6%) | 55,270 (2.7%) | 52,793 (4.0%) | 1198 (0.2%) | 5295 (3.0%) |
| **Medication use (183 days prior to four days prior)** | | | | | |
| Non-steroidal anti-inflammatory drugs | 1,259,998 (27.6%) | 515,731 (25.5%) | 355,922 (26.8%) | 147,232 (24.9%) | 48,887 (28.0%) |
| Cox2 inhibitors | 26,822 (0.6%) | 17,364 (0.9%) | 13,342 (1.0%) | 6486 (1.1%) | 1099 (0.6%) |
| Systemic corticosteroids | 255,602 (5.6%) | 128,708 (6.4%) | 99,801 (7.5%) | 32,176 (5.4%) | 10,620 (6.1%) |
| Antithrombotic and anticoagulant therapies | 110,297 (2.4%) | 70,594 (3.5%) | 58,385 (4.4%) | 16,964 (2.9%) | 5885 (3.4%) |
| Lipid modifying agents | 80,526 (1.8%) | 46,175 (2.3%) | 38,984 (2.9%) | 18,339 (3.1%) | 2736 (1.6%) |
| Antineoplastic and immunomodulating agents | 56,526 (1.2%) | 25,965 (1.3%) | 18,655 (1.4%) | 7011 (1.2%) | 3101 (1.8%) |
| Hormonal contraceptives for systemic use | 40,488 (0.9%) | 14,959 (0.7%) | 6190 (0.5%) | 2595 (0.4%) | 3087 (1.8%) |
| Tamoxifen | 1207 (0.0%) | 713 (0.0%) | 525 (0.0%) | 187 (0.0%) | 56 (0.0%) |
| Sex hormones and modulators of the genital system | 51,800 (1.1%) | 20,498 (1.0%) | 10,108 (0.8%) | 4475 (0.8%) | 3592 (2.1%) |
| One or more condition of interest* | 1,449,480 (31.7%) | 880,600 (43.6%) | 716,224 (54.0%) | 249,505 (42.1%) | 65,949 (37.8%) |
| One or more medication of interest† | 1,361,157 (29.8%) | 567,169 (28.1%) | 392,729 (29.6%) | 159,770 (27.0%) | 53,952 (30.9%) |
| One or more condition/medication of interest*† | 2,244,217 (49.1%) | 1,148,815 (56.8%) | 866,134 (65.3%) | 327,718 (55.3%) | 94,795 (54.3%) |

Characteristics of the participants in the study cohorts used for the primary analyses. Participants were aged 20 years or older and had at least one year of prior history before index date in the database. Those in the general population were present in the database as of 01/01/2017. *Conditions of interest: autoimmune disease, antiphospholipid syndrome, thrombophilia, asthma, atrial fibrillation, malignant neoplastic disease, diabetes mellitus, obesity, or renal impairment. †Medications of interest: non-steroidal anti-inflammatory drugs, Cox2 inhibitors, systemic corticosteroids, hormonal contraceptives, tamoxifen, and sex hormones and modulators of the genital system.

higher than expected among first-dose BNT162b2 recipients, with 21 observed events which compared with 12 expected (SIR: 1.70 [1.11–2.61]). These rates were more in line with expected events for second-dose of BNT162b2 and ChAdOx1. Rates of VTE with thrombocytopenia after vaccination with BNT162b2 were low and in line with expected rates. We observed 27 such events after a first-dose of BNT162b2, which compared with 21 expected (1.24 [0.85–1.82]), and 20 after a second-dose, which compared with 20 expected (0.98 [0.63–1.52]). For ChAdOx1, 8 events were seen which compared to 6 expected (1.28 [0.64–2.56]).

As for rates of ATE with thrombocytopenia, these were lower than expected with 53 events observed after a first-dose of BNT162b2, which

compared with 62 expected (0.86 [0.65–1.12]), and 59 after a second-dose, which compared with 59 expected (0.99 [0.76–1.27]). For ChAdOx1, 9 events were observed, which compared with 11 expected (0.77 [0.40–1.48]).

**Sensitivity analyses**

We found similar results in sensitivity analysis after (1) removing the requirement for at least one year of prior history available in all cohorts; (2) stratifying participants by calendar time (before March 2021 and from March 2021 onwards); and (3) excluding individuals with prior COVID-19 among those vaccinated; see Supplementary Tables 2–4. All results from primary and

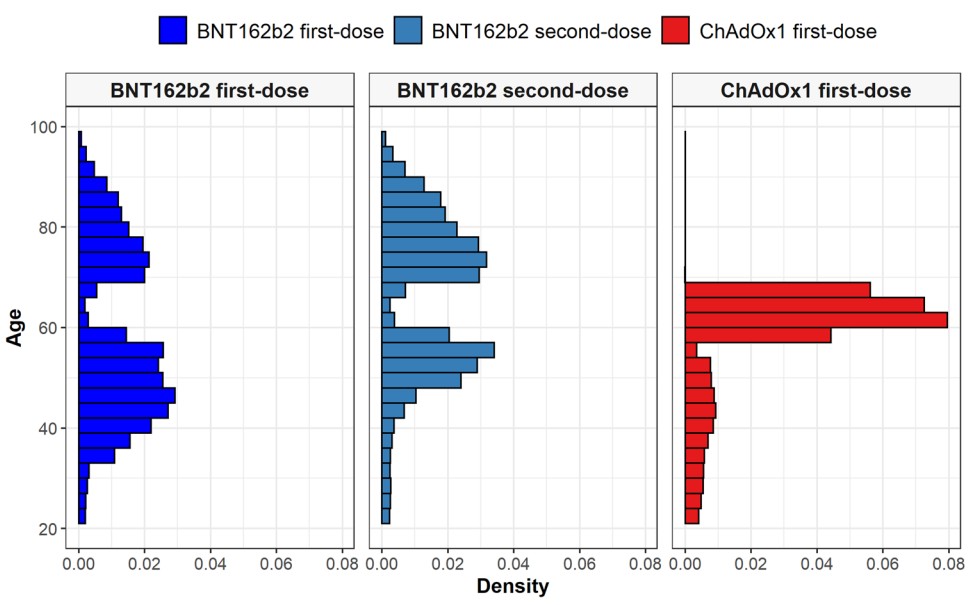

**Fig. 1 | Age profiles of people vaccinated against SARS-CoV-2.** Age distribution by vaccine and dose type. Dark blue - First-dose BNT162b2; Light blue - Second-dose BNT162b2; Red - First-dose ChAdOx1.

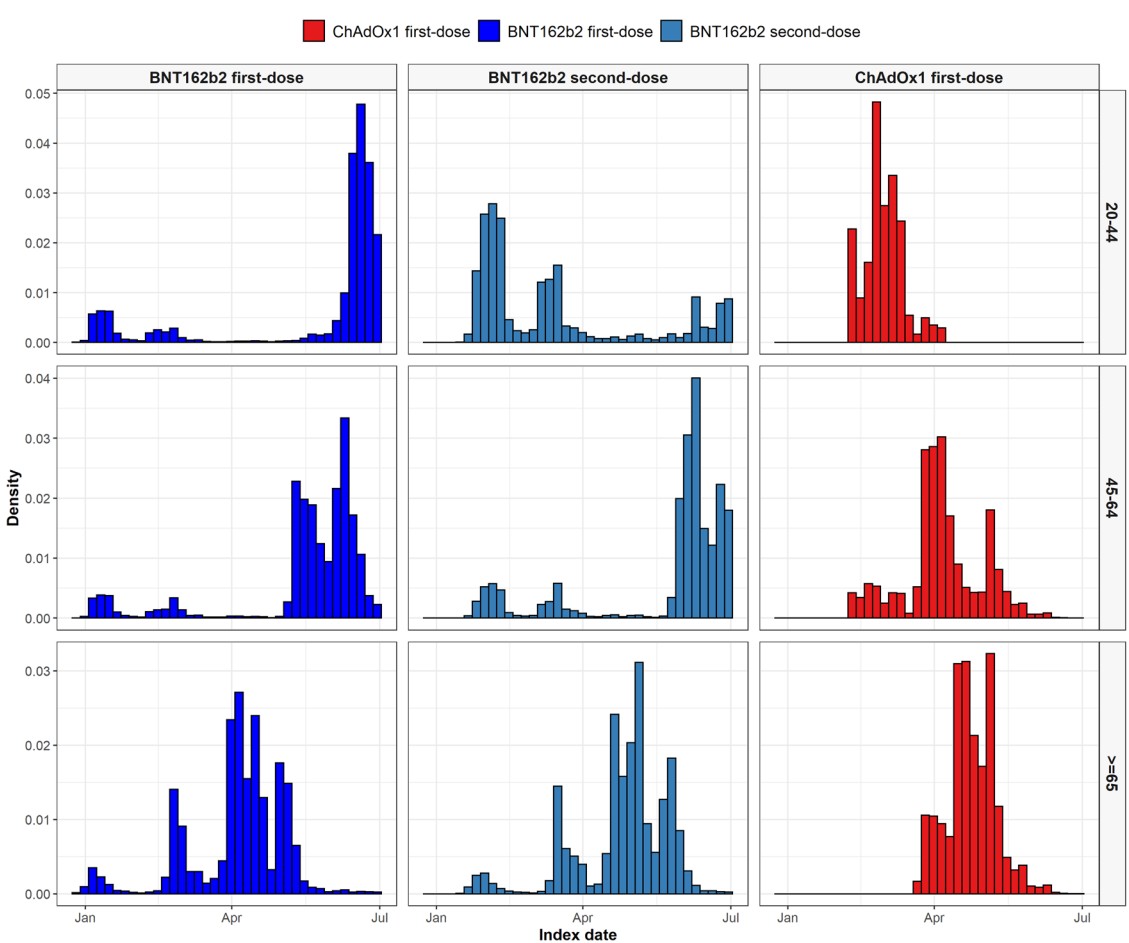

**Fig. 2 | Distribution of date of cohort entry among people vaccinated against SARS-CoV-2 by age group.** Distribution of date of cohort entry by vaccine type and age group. Dark blue - First-dose BNT162b2; Light blue - Second-dose BNT162b2; Red - First-dose ChAdOx1.

**Table 2 | Observed versus expected events among people vaccinated against SARS-CoV-2 or with a diagnosis of COVID-19**

| | N | Person-years | Observed events | Expected events | Standardised incidence ratio (95% CI) |
|---|---|---|---|---|---|
| **Deep vein thrombosis** | | | | | |
| ChAdOx1 first-dose | 590,599 | 33,928 | 39 | 43.9 | 0.89 (0.65–1.22) |
| BNT162b2 first-dose | 2,017,986 | 100,380 | 182 | 176.4 | 1.03 (0.89–1.19) |
| BNT162b2 second-dose | 1,324,426 | 78,253 | 130 | 162.6 | 0.80 (0.67–0.95) |
| COVID-19 case | 174,081 | 40,201 | 199 | 42.5 | 4.68 (4.07–5.38) |
| **Pulmonary embolism** | | | | | |
| ChAdOx1 first-dose | 590,761 | 33,938 | 24 | 30.7 | 0.78 (0.52–1.16) |
| BNT162b2 first-dose | 2,018,466 | 100,406 | 154 | 123.2 | 1.25 (1.07–1.46) |
| BNT162b2 second-dose | 1,324,890 | 78,280 | 116 | 115.8 | 1.00 (0.84–1.20) |
| COVID-19 case | 173,905 | 40,110 | 500 | 28 | 17.86 (16.37–19.50) |
| **Splanchnic vein thrombosis** | | | | | |
| ChAdOx1 first-dose | 591,119 | 33,959 | 8 | 7.3 | 1.09 (0.54–2.18) |
| BNT162b2 first-dose | 2,020,842 | 100,542 | 14 | 18.4 | 0.76 (0.45–1.29) |
| BNT162b2 second-dose | 1,326,894 | 78,401 | 9 | 16.7 | 0.54 (0.28–1.04) |
| COVID-19 case | 174,421 | 40,298 | 13 | 4.9 | 2.64 (1.53–4.55) |
| **Venous thromboembolism (deep vein thrombosis or pulmonary embolism)** | | | | | |
| ChAdOx1 first-dose | 590,307 | 33,911 | 60 | 65.5 | 0.92 (0.71–1.18) |
| BNT162b2 first-dose | 2,015,999 | 100,266 | 313 | 264.6 | 1.18 (1.06–1.32) |
| BNT162b2 second-dose | 1,322,747 | 78,151 | 227 | 246.1 | 0.92 (0.81–1.05) |
| COVID-19 case | 173,608 | 40,031 | 630 | 61.8 | 10.19 (9.43–11.02) |
| **Myocardial infarction** | | | | | |
| ChAdOx1 first-dose | 590,016 | 33,894 | 72 | 76.1 | 0.95 (0.75–1.19) |
| BNT162b2 first-dose | 2,016,950 | 100,320 | 280 | 267.3 | 1.05 (0.93–1.18) |
| BNT162b2 second-dose | 1,323,543 | 78,197 | 272 | 247.3 | 1.10 (0.98–1.24) |
| COVID-19 case | 173,975 | 40,191 | 118 | 63.7 | 1.85 (1.55–2.22) |
| **Ischaemic stroke** | | | | | |
| ChAdOx1 first-dose | 589,870 | 33,884 | 106 | 95.6 | 1.11 (0.92–1.34) |
| BNT162b2 first-dose | 2,012,434 | 100,055 | 521 | 530.5 | 0.98 (0.90–1.07) |
| BNT162b2 second-dose | 1,319,429 | 77,948 | 515 | 510.9 | 1.01 (0.92–1.10) |
| COVID-19 case | 173,390 | 39,983 | 577 | 106.5 | 5.42 (5.00–5.88) |
| **Arterial thromboembolism (myocardial infarction or ischaemic stroke)** | | | | | |
| ChAdOx1 first-dose | 588,689 | 33,815 | 178 | 168.2 | 1.06 (0.91–1.23) |
| BNT162b2 first-dose | 2,008,196 | 99,814 | 793 | 780.7 | 1.02 (0.95–1.09) |
| BNT162b2 second-dose | 1,315,777 | 77,726 | 774 | 741.9 | 1.04 (0.97–1.12) |
| COVID-19 case | 172,905 | 39,868 | 683 | 165.4 | 4.13 (3.83–4.45) |
| **Immune thrombocytopenia** | | | | | |
| ChAdOx1 first-dose | 590,919 | 33,947 | 12 | 24.9 | 0.48 (0.27–0.85) |
| BNT162b2 first-dose | 2,019,395 | 100,459 | 97 | 94.3 | 1.03 (0.84–1.26) |
| BNT162b2 second-dose | 1,325,627 | 78,325 | 61 | 89 | 0.69 (0.53–0.88) |
| COVID-19 case | 174,145 | 40,188 | 292 | 22 | 13.29 (11.85–14.91) |
| **Thrombocytopenia** | | | | | |
| ChAdOx1 first-dose | 577,534 | 33,156 | 754 | 588.5 | 1.28 (1.19–1.38) |
| BNT162b2 first-dose | 1,952,174 | 96,641 | 3186 | 2145.40 | 1.49 (1.43–1.54) |
| BNT162b2 second-dose | 1,264,798 | 74,678 | 2749 | 1963.20 | 1.40 (1.35–1.45) |
| COVID-19 case | 167,218 | 38,379 | 2476 | 539.9 | 4.59 (4.41–4.77) |
| **Deep vein thrombosis with thrombocytopenia** | | | | | |
| BNT162b2 first-dose | 2,020,823 | 100,541 | 12 | 12.8 | 0.94 (0.53–1.65) |
| BNT162b2 second-dose | 1,326,869 | 78,400 | 7 | 12 | 0.58 (0.28–1.22) |
| COVID-19 case | 174,409 | 40,295 | 25 | 3 | 8.32 (5.62–12.32) |
| **Pulmonary embolism with thrombocytopenia** | | | | | |
| BNT162b2 first-dose | 2,020,780 | 100,538 | 21 | 12.4 | 1.70 (1.11–2.61) |
| BNT162b2 second-dose | 1,326,837 | 78,398 | 15 | 11.7 | 1.29 (0.78–2.13) |
| COVID-19 case | 174,370 | 40,280 | 61 | 2.9 | 21.37 (16.63–27.46) |
| **Venous thromboembolism (deep vein thrombosis or pulmonary embolism) with thrombocytopenia** | | | | | |
| ChAdOx1 first-dose | 591,096 | 33,957 | 8 | 6.3 | 1.28 (0.64–2.56) |

**Table 2 (continued) | Observed versus expected events among people vaccinated against SARS-CoV-2 or with a diagnosis of COVID-19**

| | N | Person-years | Observed events | Expected events | Standardised incidence ratio (95% CI) |
|---|---|---|---|---|---|
| BNT162b2 first-dose | 2,020,650 | 100,531 | 27 | 21.7 | 1.24 (0.85–1.82) |
| BNT162b2 second-dose | 1,326,723 | 78,391 | 20 | 20.4 | 0.98 (0.63–1.52) |
| COVID-19 case | 174,337 | 40,272 | 78 | 5 | 15.48 (12.40–19.32) |
| Arterial thromboembolism (myocardial infarction or ischaemic stroke) with thrombocytopenia | | | | | |
| ChAdOx1 first-dose | 591,014 | 33,952 | 9 | 11.7 | 0.77 (0.40–1.48) |
| BNT162b2 first-dose | 2,020,074 | 100,497 | 53 | 62 | 0.86 (0.65–1.12) |
| BNT162b2 second-dose | 1,326,182 | 78,358 | 59 | 59.8 | 0.99 (0.76–1.27) |
| COVID-19 case | 174,297 | 40,269 | 58 | 12.6 | 4.62 (3.57–5.98) |

For each event of interest, the number of people contributing to the analysis from the target population, their person-years contributed, and the number of observed events are given. Expected events are estimated using indirect standardisation to the general population. Standardised incidence ratios (SIRs) with 95% confidence intervals (CIs) were estimated. Events with fewer than 5 occurrences were omitted to protect patient confidentiality.

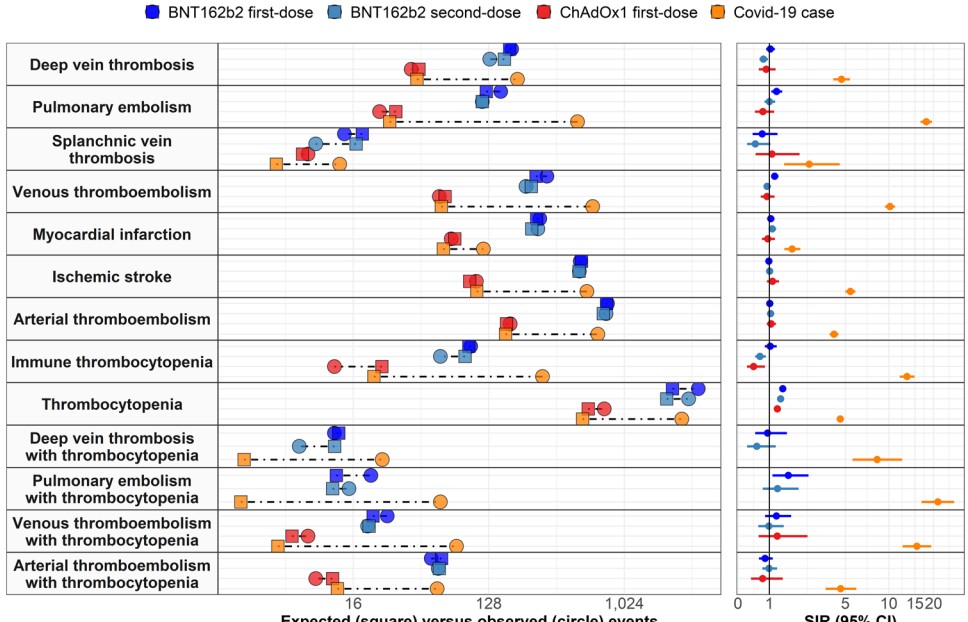

**Fig. 3 | Expected versus observed events among those vaccinated against SARS-CoV-2 and those with a COVID-19 diagnosis.** Expected events for each of the study cohorts based on indirect standardisation using rates from the general population between 2017 and 2019 are compared with the number of observed events seen in each cohort on the panels on the left. Corresponding standardised incidence ratios (SIRs) with 95% confidence intervals (95% CI) are shown in the panels on the right.

secondary analyses can be explored at: https://livedataoxford.shinyapps.io/SidiapCovidVaccinationStudy/.

## Discussion

In this study including 2,613,774 people vaccinated against SARS-CoV-2 in Catalonia, Spain, risks of thrombosis and thrombosis with thrombocytopenia after vaccination against COVID-19 were generally similar to background risks for the general population. We observed a potential safety signal for pulmonary embolism after a first dose of BNT162b2, with 31 more events than expected among 2 million vaccine recipients (SIR: 1.25 [1.07–1.46]). In addition, more cases of thrombocytopenia were seen than were expected following both first and second doses of BNT162b2 and first-dose of ChAdOx1. Rates of pulmonary embolism with concomitant thrombocytopenia were also increased among first-dose BNT162b recipients. In comparison, rates of VTE, ATE, and thrombocytopenia among COVID-19 cases were far higher than background rates. For instance, risks of VTE, ATE, and thrombocytopenia were 10-, 4- and 5-fold higher for COVID-19 cases. It is worth noting that differences exist in the socio-demographics and

clinical characteristics of recipients of both vaccines compared to each other and compared to the general population. Confounding by indication may therefore, at least in part, explain the observed safety signals.

To date, population-based studies reporting thromboembolic events among SARS-CoV-2 vaccine recipients are scarce. A cohort study from Denmark and Norway which included 281,264 people aged between 18 and 65 years vaccinated with ChAdOx1 reported a 2-fold increase in VTE (SIR: 1.97 [1.50–2.54]) and a 3-fold increase in thrombocytopenia (SIR: 3.02 [1.76–4.83]) within 28 days of vaccination[11]. VTE rates were largely driven by cerebral venous sinus thrombosis (CVST) events, and were higher among younger age groups (18-44 years SIR: 2.99 [1.94–4.42]). Further, in line with our results, Pottegård et al. did not observe increased rates of ATE. Differences between ours and Pottegård's results might be partially explained by the characteristics of ChAdOx1 vaccine recipients, who were mostly aged 60–69 years in our study.

In a nested case-control study including hospital data from Scotland, increased rates of idiopathic thrombocytopenic purpura (ITP),

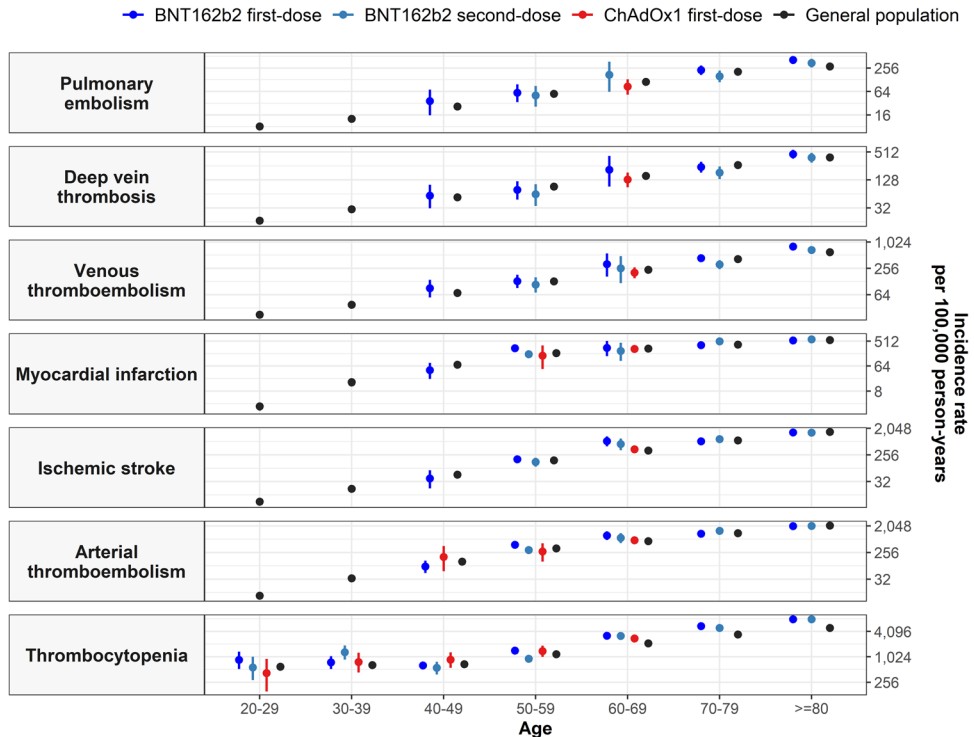

**Fig. 4 | Background and post-vaccine rates of thromboembolic events and thrombocytopenia by age.** Events with less than 5 occurrences have been omitted for privacy reasons. Point estimates with 95% confidence intervals.

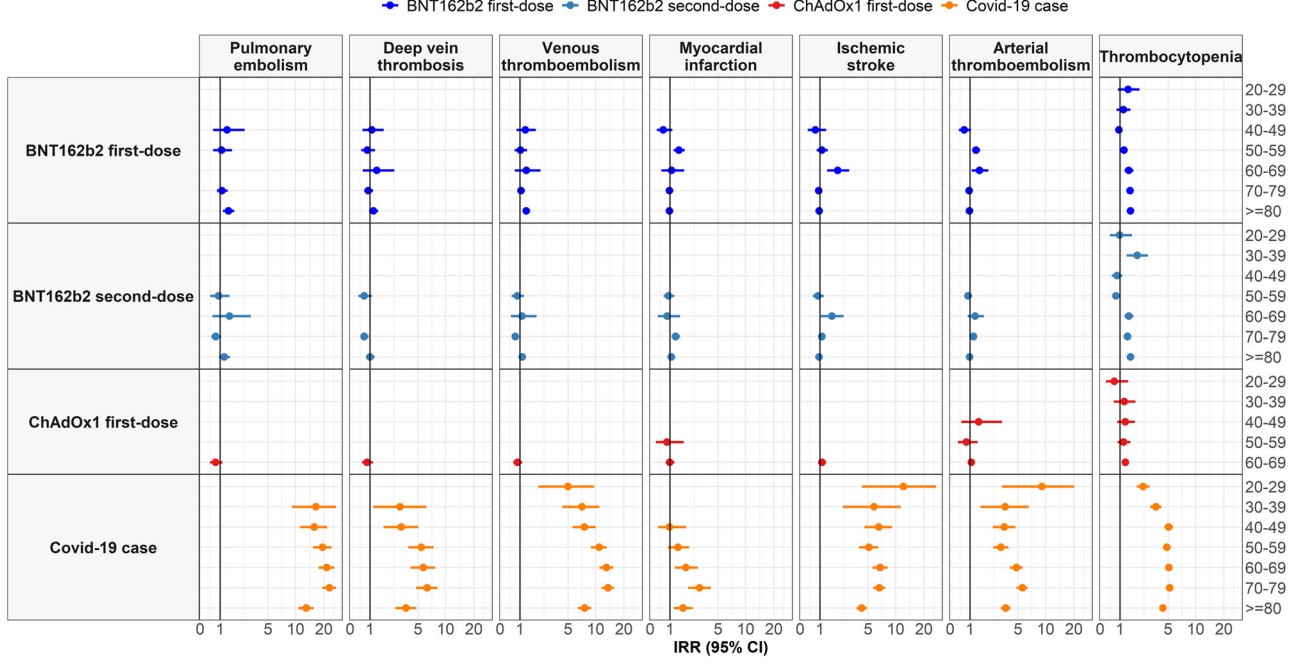

**Fig. 5 | Incidence rate ratios (IRRs) for thromboembolic events and thrombocytopenia by age.** Events with less than 5 occurrences have been omitted for privacy reasons. Point estimates with 95% confidence intervals.

ATE and haemorrhagic events were observed among 1.7 million people with a first-dose of ChAdOx1[12]. These findings were subsequently confirmed for ITP in a post hoc self-controlled case series (SCCS) analysis, but not for ATE or haemorrhagic events. In contrast to our results, the authors did not find increased rates of VTE among 821,052 people vaccinated with a first-dose of BNT162b2. Discrepancies between their and our results could be related to the study design, as

well as to different sample sizes. As recently raised by Schuemie et al., case-control studies are prone to substantial bias when using retrospective databases and therefore must be interpreted with caution[14]. Further, our study included a larger cohort of BNT162b recipients but fewer and older ChAdOx1 vaccinees.

Another SCCS study from the UK found increased risks of hospital admissions or deaths associated with thromboembolism events

among vaccine recipients and individuals with SARS-CoV-2[15]. ChAdOx1, for which 19.6 million recipients were included, was associated with increased risks of thrombocytopenia, VTE, and CVST 8–14 days after first-dose vaccination, while BNT162b, with 9.5 million recipients, was associated with increased risks of ATE, ischaemic stroke, and CVST 15–21 days after first-dose vaccination. Interestingly, although we did not find an association between BT162b and ATE overall, we observed increased risks of ATE among BNT162b2 recipients aged 50-69 years. As also seen in our study, risks for these events were much higher among individuals with SARS-CoV-2. Again, sample sizes and the characteristics of ChAdOx1 recipients differed from ours, and only severe events leading to hospitalisation or death were captured in their study.

In a cohort study conducted by our group using primary care data from the UK, we also found increased risks of pulmonary embolism (SIR: 1.21 [1.07–1.36]) among 1.6 million BNT162b first-dose recipients[16]. Risks of pulmonary embolism were also seen to be increased among 1.8 million ChAdOx1 recipients. Rates of thrombocytopenia were also higher than expected for ChAdOx1, with similar SIR to those in Catalonia (SIR 1.25 [1.19–1.31]), but not for BNT162b recipients. However, rates of immune thrombocytopenia were higher than expected for both vaccines, with SIRs of 2.01 [1.27–3.19] for ChAdOx1 and 1.74 [1.05–2.89] for BNT162b2. Likewise, the differences between the two studies might be related to differences in study populations, with the UK study including younger and three-times as many ChAdOx1 recipients, but fewer BNT162b vaccinees. Differences could also be related to lack of hospital data in the UK study.

The mechanisms underlying a potential association between SARS-CoV-2 vaccines and thromboembolic events are currently under investigation. Shortly after the first signal alerts for the ChAdOx1 vaccine, case reports were published describing a potentially new immune disorder named vaccine-induced immune thrombotic thrombocytopenia (VITT) among ChadOx1 recipients[17,18]. According to the author's descriptions, this disorder manifests with atypical thrombotic events (e.g. CVST, SVT) associated with thrombocytopenia 5 to 15 days following vaccination and could be mediated by platelet-activating autoantibodies against platelet factor 4 (PF4). Regarding SARS-CoV-2 mRNA vaccines, some authors have suggested that the inflammatory response following vaccination might increase macrophage-mediated clearance and/or diminish platelet production, thus leading to thrombocytopenia[13]. Such mechanisms have been previously postulated in relation to ITP following viral infections (including SARS-CoV-2)[19], as well as following vaccination against other virus (e.g., varicella-zoster, measles-mumps-rubella)[20,21]. We did find increased risks of pulmonary embolism with thrombocytopenia among BNT162b2 recipients and increased rates of thrombocytopenia among BNT162b2 and ChadOx1 recipients. However, risks of thrombocytopenia and TTS were far higher among COVID-19 cases. Further research is needed to confirm our observations in other large population-based cohorts and to establish its pathogenesis.

The main strength of our study is its large sample size and representativeness, which allowed us to assess the incidence of rare adverse events in a real-world setting. In addition, to our knowledge, this is the largest study to-date reporting thromboembolic events following BNT162b2 second-dose. Our study was underpinned by a well-established primary care database that has previously been used for various post-authorisation safety studies[22,23], linked to hospital diagnoses, and was designed in collaboration with the study funder, the EMA. Finally, for the sake of transparency and reproducibility, we have made our protocol, analytical code and full result set publicly available.

However, our study also has several limitations. First, vaccinated cohorts differed substantially from the general population and from one another. Our data mirrored the different, and changing, nation-wide guidelines for the provision of BNT162b2 and ChAdOx1 in Spain.

Although the distribution of comorbidities and medication was broadly similar across cohorts when stratifying by age categories, vaccine recipients were generally in slightly worse health than the general population. Therefore, even though we calculated SIRs to allow comparisons between cohorts, we cannot exclude that residual confounding by indication might have influenced our results. In addition, the majority of ChAdOx1 vaccine recipients were older than 60 years in our study, with prior evidence suggesting a stronger association between this vaccine and VTE among individuals younger than 50 years[11,15]. This could have prevented us from observing an association between ChAdOx1 and VTE. Secondly, thromboembolic events were identified using routinely collected data, and therefore the lack of formal adjudication of outcomes might have introduced measurement error. Events might also be underestimated in all cohorts, especially rare events with complex diagnoses, such as immune thrombocytopenia or CVST, which we were unable to identify. Although underestimation of events was likely non-differential across cohorts for events before March 2021, we cannot exclude that detection bias following the EMA's signal alert report might have led to differential measurement error between the vaccinated and comparator cohorts from March onwards. However, our results were consistent in analysis stratified by time period for the first-dose BNT162b2 and ChadOx1 cohorts. Thirdly, individuals with a thromboembolic event after a first-dose were by design excluded from an analysis of the same event after a second-dose and this exclusion could be considered as leading to depletion of susceptibles for the second-dose cohort.

In summary, in this study of over two million people vaccinated against SARS-CoV-2 in Catalonia, Spain, the BNT162b2 and ChAdOx1 vaccines have been seen to have similar safety profiles. Safety signals for pulmonary embolism following BNT162b vaccination, as well as for thrombocytopenia following BNT162b2 and ChAdOx1 vaccination have been identified. Our results must be interpreted with caution considering the risk of residual confounding by indication. Regardless of the vaccine used, the increase in rates of thrombosis among persons with COVID-19 is far higher than any potential safety signal seen among persons vaccinated.

## Methods
### Study design and data source
We conducted a population-based cohort study using data from the Information System for Research in Primary Care (SIDIAP; www.sidiap. org), a primary care database from Catalonia. This study was approved by the Clinical Research Ethics Committee of the IDIAPJGol (project code: 21/054-PCV). SIDIAP includes pseudo-anonymized primary care electronic health records from 80% of the population in Catalonia and is representative of the general population in terms of age, sex, and geographic distribution[24] SIDIAP was linked to discharge data from hospitals in Catalonia, with this data including both diagnosis and procedures registered during hospital admissions. The database has been mapped to the Observational Medical Outcomes Partnership (OMOP) Common Data Model (CDM), a structure that facilitates safety surveillance in observational health care databases[25].

### Setting
The national COVID-19 vaccination campaign was launched in Spain on 27 December 2020. Spain established priority subpopulation groups eligible for vaccination, differentiating the general population and essential workers[26]. Among the general population, vaccination began with those considered most at risk of severe disease (people living in nursing homes, older people, and people with risk factors for COVID-19), whereas among essential workers vaccination began with health-care professionals followed by others (such as teachers and police officers). While the BNT162b2 vaccine has consistently been used across both of these priority groups, guidelines for ChAdOx1 have

changed over time. For instance, ChAdOx1 was initially restricted to essential workers aged 55 years or younger, but its use became restricted to those aged 60 to 65 and, subsequently, from 60 to 69 years after safety concerns emerged in March. Of note, we did not include mRNA-1273, Ad.26.COV2.S and second-dose ChAdOx1 recipients in this study due to small sample sizes.

### Study participants and follow-up time
Three vaccination cohorts were identified: BNT162b2 first-dose, BNT162b2 second-dose, and ChAdOx1 first-dose. Individuals vaccinated were required to have received their vaccine between 27 December 2020 and 23 June 2021 (a week prior to the end of data availability), be aged ≥20 years at time of vaccination (with very few people under 20 vaccinated during the study period), and have at least a year of prior history available (so as to identify events of interest prior to vaccination). In addition, those with a second-dose of BNT162b2 were required to have received this dose between 18 to 28 days following their first.

A historical comparator general population cohort was identified. Individuals present in the database as of 1 January 2017 were included in this cohort, with this date used as their index date. In addition, a cohort of COVID-19 cases was identified between 1 September 2020 and 23 June 2021 (i.e. from the second-wave onwards in Spain). These individuals were identified on the basis of a positive PCR test for SARS-CoV-2. Only incident cases were included, with persons with a clinical diagnosis of COVID-19 or positive test for SARS-CoV-2 prior to 1 September 2020 excluded from this cohort and only the first-recorded positive PCR test used for those persons included. COVID-19 cases were also required to have not had a vaccination against COVID-19 prior to their index date. As with the cohorts of people vaccinated, individuals were required to be aged ≥20 years and have at least a year of prior history available.

Additional cohorts were generated for sensitivity analyses. We created analogous cohorts including all individuals aged ≥20 years with no requirement for the amount of prior history available. Subsequently, we excluded any individuals with a COVID-19 diagnosis prior to the index date for the vaccinated cohorts. Vaccination cohorts were also stratified by calendar time; before 1 March 2021 and from 1 March 2021 onwards.

For each specific outcome and cohort, we excluded individuals with an occurrence of the outcome the year prior to the index date. All cohorts were followed from index date to whichever came first of: end of follow-up (21 days for people vaccinated, 90 days for those diagnosed with COVID-19, and 31 December 2019 for the background general population cohort), end of data collection (26 May 2021), exit from the database, or occurrence of the outcome of interest. In addition, follow-up was censored at time of second-dose for the BNT162b2 and ChAdOx1 first-dose in the few cases where it occurred before 21 days.

### Outcomes
Venous thromboembolic events included deep vein thrombosis (DVT), pulmonary embolism (PE), and the composite venous thromboembolism (VTE, which included DVT and PE). We also assessed portal vein thrombosis and splanchnic venous thrombosis (SVT). Arterial thromboembolism (ATE) events included myocardial infarction and ischaemic stroke. We also identified stroke in general, for which we included both ischemic and haemorrhagic stroke. Thrombocytopenia was identified using diagnostic codes or a measurement of a platelet count between 10,000 and 150,000 platelets/microliter. Finally, thromboembolic events with concomitant thrombocytopenia were identified, where thrombocytopenia was seen in 10 days before and after the thromboembolic event.

### Statistical analyses
We first summarised the characteristics of individuals included in each cohort (socio-demographics, baseline comorbidities and drug prescriptions), with counts and percentages for categorical variables and median and interquartile ranges (IQR) for continuous variables. For each cohort and outcome, we described the total number of events observed and calculated incidence rates (IR) per 100,000 person-years, with exact 95% confidence intervals (CI), overall and stratified by age and sex. We calculated crude incidence rate ratios (IRRs), with 95% CI, for the vaccinated and COVID-19 cohorts compared against the background general population cohort, both overall and stratified by age and sex. We also estimated the number of events expected among the vaccinated and COVID-19 cohorts using indirect standardisation (10 year age bands), with the general population cohort as the standard population. We calculated standardised incidence ratios (SIRs) with 95% CI dividing the number of events observed by the number of events expected. A SIR above 1 indicates that the observed rate for a specific outcome was higher than what was expected in the population and was taken to indicate a safety signal for a given vaccine cohort.

### Reporting summary
Further information on research design is available in the Nature Portfolio Reporting Summary linked to this article.

## Data availability
In accordance with current European and national law, the data used in this study is only available for the researchers participating in this study. Thus, we are not allowed to distribute or make publicly available the data to other parties. However, researchers from public institutions can request data from SIDIAP if they comply with certain requirements. Further information is available online (https://www.sidiap.org) or by contacting SIDIAP (sidiap@idiapjgol.info). Source data are provided with this paper.

## Code availability
The analytic code to perform the study is available at https://github.com/SIDIAP/CovidVaccinationAdverseEventsStudy (https://doi.org/10.5281/zenodo.6583871).

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

## Acknowledgements
This study was funded by the European Medicines Agency in the form of a competitive tender (Lot ROC No EMA/2017/09/PE). This document expresses the opinion of the authors of the paper, and may not be understood or quoted as being made on behalf of or reflecting the position of the European Medicines Agency or one of its committees or working parties. E.R. was supported by Instituto de Salud Carlos III (grant number CM20/00174) and conducted this study as part of the doctoral program in methodology of biomedical research and public health at the Autonomous University of Barcelona. D.P.A. is funded through a National Institute for Health Research (NIHR) Senior Research Fellowship (Grant number SRF-2018-11-ST2-004).

## Author contributions
E.B., X.L., V.Y.S., D.P.A. and T.D.S. led study design. E.R., A.P., S.F.B., B.R., M.A. and T.D.S. led data collection and processing and analysis. K.V. and C.R. provided clinical input and contributed to the identification of study outcomes. P.R. led the coordination of the project and contracting. E.B., E.R., and D.P.A. led the drafting of the manuscript. All authors were involved in the interpretation of the results, and the critical review and approval of the manuscript.

## Competing interests
D.P.A.'s research group has received research grants from the European Medicines Agency, from the Innovative Medicines Initiative, from Amgen, Chiesi, and from UCB Biopharma; and consultancy or speaker fees from Astellas, Amgen and UCB Biopharma. The remaining authors declare no competing interests'

## Additional information

**Edward Burn** ⓘ[1,2,5], **Elena Roel** ⓘ[1,3,5], **Andrea Pistillo** ⓘ[1], **Sergio Fernández-Bertolín**[1], **Maria Aragón** ⓘ[1], **Berta Raventós**[1,3], **Carlen Reyes**[1], **Katia Verhamme**[4], **Peter Rijnbeek** ⓘ[4], **Xintong Li** ⓘ[2], **Victoria Y. Strauss**[2], **Daniel Prieto-Alhambra** ⓘ[2,4,6] ✉ & **Talita Duarte-Salles** ⓘ[1,6] ✉

[1]Fundació Institut Universitari per a la recerca a l'Atenció Primària de Salut Jordi Gol i Gurina (IDIAPJGol), Barcelona, Spain. [2]Centre for Statistics in Medicine (CSM), Nuffield Department of Orthopaedics, Rheumatology and Musculoskeletal Sciences (NDORMS), University of Oxford, Oxford, UK. [3]Universitat Autònoma de Barcelona, Bellaterra (Cerdanyola del Vallès), Barcelona, Spain. [4]Department of Medical Informatics, Erasmus University Medical Center, Rotterdam, The Netherlands. [5]These authors contributed equally: Edward Burn, Elena Roel. [4]These authors jointly supervised this work: Daniel Prieto-Alhambra, Talita Duarte-Salles. ✉e-mail: daniel.prietoalhambra@ndorms.ox.ac.uk; tduarte@idiapjgol.org

