## [Peer review file · Nature Communications]

Title: Thrombosis and thrombocytopenia after vaccination against and infection with SARS-CoV-2 in Catalonia, SpainREVIEWER COMMENTS

Reviewer #1 (Remarks to the Author): 
I wonder it is acceptable to publish similar types of articles written by the same first author independently. It may be better the second article follows the other and refers to the first article.

Reviewer #2 (Remarks to the Author):

Burns et al. present here a further analysis of primary care data from Spain seeking to measure incidences of VTE, ATE other thromboses and / or thrombocytopenia in temporal relationship with SARS-COV-2 vaccinations. The weaknesses of the back to back submission from the UK data set apply here as well, because the major proportion of vaccinees is older, which might miss a safety signal of TTS.

However, I suggest to provide an analysis of a combined measure involving 1. any thrombosis (arterial+venous) and 2. thrombocytopenia. These risks should be compared for different age strata (12-30, 30-60, >60).

Reviewer #3 (Remarks to the Author): 
See attachment

Reviewer #1 (Remarks to the Author):

I wonder it is acceptable to publish similar types of articles written by the same first author independently. It may be better the second article follows the other and refers to the first article.

We thank the reviewer for their suggestion. We have updated the discussion section with references to the preprint of the study conducted in the United Kingdom (lines 224-234).

I appreciate another opportunity to review an interesting article. This is the second article from the same first author. Another similar study (NCOMMS-21-33049-T) was done in a different cohort with a similar method. This reviewer agrees that the thrombotic side-effects after mRNA vaccine is a current major topic and this type of study provides important information to the physicians as well as the developers of new vaccines.

1. As this reviewer had asked before (in the first paper), the increased risk of thrombosis after BNT162b2 vaccination has not been emphasized. How severe were they? How many required ICU treatments?

We agree with the reviewer that providing information regarding the degree of severity of thromboembolic events would be of great interest. Unfortunately, due to the nature of our data, we do not have such information. This population-based study was underpinned by the SIDIAP database, which includes pseudo anonymised primary care electronic health records covering 80% of the population in Catalonia. While our database provides a comprehensive overview of demographics, medical diagnoses and drug prescriptions, we do not have information regarding the symptoms or severity of these events. Similarly, although we have updated our results in order to include hospital data, we only have information regarding diagnoses made at hospitals, without any information regarding severity nor intensive services requirements.

2. How did you screen DVT? Were all the patients symptomatic? Was the diagnosis was made by Doppler echo or venography? Were all the DVT patients examined for pulmonary embolism? Do all the facilities follow the common strategy for thrombosis?

This study was underpinned by electronic health records, and so it includes data routinely collected by healthcare professionals during their daily practice. Therefore, diagnosis of DVT (and other events)

were made at each facility on the basis of standard practice. Further, as mentioned before, due to the nature of our data we do not have information regarding symptomatology nor regarding how the diagnoses were made. While we acknowledge that including outcomes without formal adjudication might introduce a measurement error bias, we believe that this would be non-differential across cohorts, aside for events after March 2021. Indeed, increased surveillance following the EMA's signal alert report could have introduced differential bias in our results. However, when stratifying our results by time period (before or after March) our results were consistent with the main analysis.

3. The mystery in this report is the incidence of thrombocytopenia which was more than expected after BNT162b2 vaccination. Furthermore, the incidence did not increase after the ChAdOx1 vaccination. These observations are contradictory to the results in the first report. If an insufficient number of cases is the reason for these inconsistent results, that may affect the evaluation of the reports.

After updating our results (including more vaccine recipients and hospital diagnoses of thromboembolic events), we have observed increased risks of thrombocytopenia among BNT162b2 (SIR: 1.49 [1.43 to 1.54] and 1.40 [1.35 to 1.45]) for first and second doses, respectively) and ChAdOx1 recipients (1.28 [1.19 to 1.38]).

Prior observational studies have also reported increased risks of thrombocytopenia among ChAdOx1 recipients.^{1,2} This is also in line with our findings in the UK study, with a SIR of thrombocytopenia among ChAdOx1 recipients of 1.25 [1.19 to 1.31]).³ Conversely, in the UK we did not find an association between BNT162b2 and thrombocytopenia, but we did observe increased risks of immune thrombocytopenia among both ChAdOx1 and BNT162b2 recipients (SIRs of 2.01 [1.27 to 3.19] and 1.74 [1.05 to 2.89], respectively), as well as SARS-CoV-2 cases. These differences could be explained by differences in sample size (with more BNT162b2 vaccinees in the Spanish study), different coding practices, as well as the lack of hospital data in the UK study.

1. Pottegård, A. *et al.* Arterial events, venous thromboembolism, thrombocytopenia, and bleeding after vaccination with Oxford–AstraZeneca ChAdOx1-S in Denmark and Norway: Population based cohort study. *BMJ* **373**, (2021).
2. Hippisley-Cox, J. *et al.* Risk of thrombocytopenia and thromboembolism after covid-19 vaccination and SARS-CoV-2 positive testing: Self-controlled case series study. *BMJ* **374**, (2021).

3. Burn, E. *et al.* Thrombosis and thrombocytopenia after vaccination against and infection with SARS-CoV-2: a population-based cohort analysis. *medRxiv* 2021.07.29.21261348 (2021) doi:10.1101/2021.07.29.21261348.

4. The definitive diagnosis of immune thrombocytopenia is not easy. The clinical diagnosis is usually made by ruling out other possible causes. In this study, did you rule out VITT by measuring anti-PF4 antibody/polyanion antibody by ELISA or platelet functional tests?

As previously discussed, we analysed data obtained from routinely-collected health records at the primary care and hospital level. Therefore, diagnoses of thrombocytopenia, as well as immune thrombocytopenia, were made at the hospital level as part of routine practice following standard procedures at each facility.

5. The description in the Summary part, “BNT162b2 and ChAdOx1 vaccines have been seen to have similar safety profiles. Safety signals for both venous thromboembolic events and thrombocytopenia following BNT162b2 vaccination have been identified, with their magnitude similar to these same events among people vaccinated with ChAdOx1” sounds odd to me. Since VITT is generally thought to occur almost only after ChAdOx1 vaccination and its mortality is very high, the risk/benefit is strictly evaluated in the use of this vaccine. Don’t you think it might not possible to identify the risk of this vaccine correctly in this study?

After updating our results in order to include more vaccine recipients and hospital diagnoses, we found increased risks of thrombocytopenia among BNT162b2 and ChAdOx1 recipients, as well as increased risks of pulmonary embolism and pulmonary embolism with thrombocytopenia among first dose BNT162b2 recipients.

Other observational studies have also reported increased risks of arterial and venous thromboembolism events following COVID-19 vaccination with ChAdOx1/BNT162b2.^{1,2,3,4} As highlighted in the discussion section (lines 192-223), studies (some of which have been published since we submitted the original version of our manuscript) have reported increased risks of VTE among ChAdOx1^{1,2} and BNT162b2 recipients,³ increased risks of ATE among BNT162b2 recipients,² increased risks of thrombocytopenia among ChAdOx1 recipients^{1,2,3}, and increased risks of immune thrombocytopenia among ChAdOx1⁴ and BNT162b2 recipients.³ Thus, thromboembolic events under investigation in relation to COVID-19 vaccines are not limited to vaccine-induced immune thrombotic thrombocytopenia (VITT).

Our data reflects the way in which vaccines have been given in Spain. As a result our ChAdOx1 population does have a very particular age distribution, with the vast majority of recipients aged between 60 and 69. We have highlighted this in Figure 1 and the results section and we agree that this may be a key reason we did not find the increased risk for VTE that has been reported elsewhere. We have noted this in the discussion when putting results into context (lines 192 to 234) and noted this as a limitation of the study (lines 268 to 271).

1. Pottegård, A. *et al.* Arterial events, venous thromboembolism, thrombocytopenia, and bleeding after vaccination with Oxford-AstraZeneca ChAdOx1-S in Denmark and Norway: Population based cohort study. *BMJ* **373**, (2021).
2. Hippisley-Cox, J. *et al.* Risk of thrombocytopenia and thromboembolism after covid-19 vaccination and SARS-CoV-2 positive testing: Self-controlled case series study. *BMJ* **374**, (2021).
3. Simpson, C. R. *et al.* First-dose ChAdOx1 and BNT162b2 COVID-19 vaccines and thrombocytopenic, thromboembolic and hemorrhagic events in Scotland. *Nat. Med.* **27**, 1290–1297 (2021).
4. Burn, E. *et al.* Thrombosis and thrombocytopenia after vaccination against and infection with SARS-CoV-2: a population-based cohort analysis. *medRxiv* 2021.07.29.21261348 (2021) doi:10.1101/2021.07.29.21261348.

Reviewer #2 (Remarks to the Author):

Burns et al. present here a further analysis of primary care data from Spain seeking to measure incidences of VTE, ATE, other thromboses and / or thrombocytopenia in temporal relationship with SARS-COV-2 vaccinations. The weaknesses of the back to back submission from the UK data set apply here as well, because the major proportion of vaccinees is older, which might miss a safety signal of TTS.

However, I suggest providing an analysis of a combined measure involving 1. any thrombosis (arterial+venous) and 2. thrombocytopenia. These risks should be compared for different age strata (12-30, 30-60, >60).

We thank the reviewer for this suggestion. However, our analyses were done following a pre-specified protocol designed in collaboration with our study funder, the European Medicines Agency (EMA). The protocol for the study was registered on EU PAS (Register Number EUPAS40414) and in this protocol age groups and outcomes were all pre-specified.

When designing the study we chose to assess arterial and venous thrombotic events separately because the mechanisms underlying arterial and venous thrombosis are substantially different.¹ Indeed, the majority of cases of arterial thrombosis are due to the rupture of an atherosclerotic plaque in an artery, which leads to platelet aggregation and thrombus formation. Conversely, venous thrombosis is related to endothelial dysfunction, hypercoagulability, and blood stasis in veins. Arterial thrombi are rich in platelets, while venous thrombi are rich in fibrin and red blood cells. In addition, treatments for these events are also substantially different, with arterial and venous thrombosis being fundamentally treated with drugs targeting platelets and the coagulation cascade, respectively.² For those reasons, in our discussions with the EMA, it was decided that we would analyse the risks of arterial and venous thrombosis separately.

1. Koupnova, M., Kehrel, B. E., Corkrey, H. A. & Freedman, J. E. Thrombosis and platelets: an update. *Eur. Heart J.* **38**, 785–791 (2017).
2. Mackman, N. Triggers, targets and treatments for thrombosis. *Nature* **451**, 914 (2008).

Reviewer #3 (Remarks to the Author):

This large cohort study "Thromboembolic events and thrombosis with thrombocytopenia after COVID-19 infection and vaccination in Catalonia, Spain" by Edward Burn et al. compares incidence of thrombosis and thrombocytopenia after vaccination with BNT162b2 (first and second dose, BNT162b2 (manufactured by Pfizer-BioNtech or Moderna) and ChAdOx1 (first dose, (AstraZeneca or Janssen) with the rates among the general population before the COVID-19 82 pandemic. These results are very important because the media and even public health institutions made big news about such events among vaccinated persons without giving any scale whether the incidence is at a normal level or higher. Such information may have decreased vaccination interest and slowed down the speed of the fight against the Covid-19 pandemic. Several European countries paused ChAdOx1 vaccinations after spontaneous reports of unusual thromboembolic events associated with thrombocytopenia among ChAdOx1 recipients.

This cohort study has an excellent design using data from a primary care database from Catalonia. The authors refer to their main outcome measure as the "standardised incidence rate ratio", but apparently it is the standardised incidence ratio (SIR), i.e., the ratio of observed and expected numbers of events. It is not clearly described how the person-years and the expected numbers were calculated; this should be clarified.

We thank the reviewer for their comments. We did calculate Standardised Incidence Ratios (SIR), and have clarified this in the manuscript. We have also added in the Methods more details explaining how SIRs were calculated (lines 436-446)

The results stratified by broad age categories and sex are expressed as crude incidence rate ratios (IRRs). The authors should explain why all results could not be given as SIRs, with proper adjustment for age (which is crucial in this study where the sub-cohorts are very different in their age distributions). It would also make the paper more reader-friendly if all results would be based on the same clearly defined measure.

We thank the reviewer for this suggestion. We have used SIRs to provide an overall summary measure for each vaccine which takes into account the differences in age distributions of those vaccinated and the general population. We have though now also updated the paper with greater emphasis on the results within the narrower, 10 year age groups (see updated Figure 4 and 5 for example) rather than the broader age groups that we previously focused on.

The reference incidence rates are taken from the period of the three first weeks of the year 2017, while the other cohorts were followed-up from late 2020 to early 2021. It would be important to understand whether there are time trends or seasonal variation in the incidence of the outcome events in Catalonia. If not, the authors should tell this in the paper. If there is temporal variation, this should be corrected in the analyses.

To clarify, we compared incidence rates among individuals vaccinated and/or infected with SARS-CoV-2 to incidence rates between the 1st January 2017 and 31st December 2019. This allowed for a sufficient sample size and person years of follow up to estimate background rates. While restricting background rates to the same calendar months as those being vaccinated (December to June) may have helped to have alleviated any potential issues due to seasonality, this would have substantially reduced the person-years contributed and increased the uncertainty in background rates.

In the first 21 days following the first dose of BNT162b2, there was a 15-29% excess of deep vein thrombosis (DVT), pulmonary embolism (PE) and venous thromboembolism (VTE). After the second dose of BNT162b2, the observed numbers were 10-13% lower than expected numbers. For the first-dose ChAdOx1 cohort, the observed and expected numbers of DVT and PE were almost identical. In comparison, there was an 8-fold excess in the cohort of persons with COVID-19 infection. Since the outcomes are very rare, it would be good to also give an estimate of absolute excess risk. E.g., the expression “1.3 fold increase in the rate of VTE” in the last sentence of the Abstract might be written in a less frightening way: “one excess cases per 25,000 vaccinated persons”.

The discussion gives the impression that potential biases are small and, if there are any, they tend to raise the risk estimates in the vaccinated cohorts. It would be good to highlight in the conclusion and Abstract that these are rather too high than too low.

We are thankful to the Reviewer for highlighting this issue. We agree that the risk of residual bias is high given the nature of our study and have highlighted this in the Conclusion and in the Abstract. For the same reason, we decided not to provide estimates of absolute excess risks, since we believe that such estimates might be interpreted as if there was a definitive causal relationship between COVID-19 vaccine and thromboembolic events. We have though provided examples of the numbers of expected versus observed events, for example “We observed a potential safety signal for pulmonary embolism after a first dose of BNT162b2, with 31 more events than expected among 2 million vaccine recipients”. We hope this helps to contextualise the results and underline how rare these events were.

Minor:

The manuscript has apparently been written in a rush (which is understandable with this topic) and therefore there are several small details which could be improved. E.g., confidence intervals in one paragraph are given in three different styles: "1.94 to 4.42", "1.81-3.14" and "0.83 to 1-58".

We thank the reviewer for catching these mistakes, which have been corrected in the updated manuscript.

Figure 3. The scale in y axis for the SIR is wrong; the SIR cannot have negative values. The graphical presentation of the observed and expected numbers of events is messy. Same information is given in Table 2 and therefore Figure 3 can be dropped. If the authors/journal prefers graphical presentation, a graph with only SIRs and CIs would be OK.

We have updated Figure 3 and corrected the axis. We agree that this figure presents a lot of results, but we have kept both panels as we believe they help to show both relative and absolute risks. These numbers are also shown in Table 2, but we hope readers will find this a useful overview of the results from the study.

Figure 4. Far too much information in one figure. The short y axes make it difficult to see whether there are differences between the categories. The values shown on the y axis (e.g., 8-64-512 or 512-2048-8192; same height for 64- and 16-fold differences) make it challenging to estimate the actual incidence rates of the dots marked in the graph between the values. If these results are important, they should be presented in a Table.

We agree that the prior figure included a lot of information. We have now updated this figure in order to include incidence rates for both sexes together. While we agree that using different scales on the y axis makes the interpretation challenging, the main message of this figure is to see how incidence rates of these events increase with age. Thus, we believe that the Figure conveys this message more clearly than a table.

Figures 5-6. The x axis should be logarithmic because the graph is intended to illustrate relative differences (IRR). In the current graphs, only values of >1 are visible while values of <1 are not.

We have replaced these two figures with a single new figure, Figure 5. As suggested we have used logarithmic scales which we agree improve legibility.

REVIEWERS' COMMENTS

Reviewer #1 (Remarks to the Author):

A simple summary table that compares the results from UK and Spain studies will help the understanding.

Reviewer #2 (Remarks to the Author):

I have no additional comments.

Reviewer #3 (Remarks to the Author):

The authors have reacted to the reviewer's suggestions in a way that makes the manuscript acceptable.